# Is It Correct to Consider Caustic Ingestion as a Nonviolent Method of Suicide? A Retrospective Analysis and Psychological Considerations

**DOI:** 10.3390/ijerph20136270

**Published:** 2023-06-30

**Authors:** Rosa Gravagnuolo, Stefano Tambuzzi, Guendalina Gentile, Michele Boracchi, Franca Crippa, Fabio Madeddu, Riccardo Zoja, Raffaella Calati

**Affiliations:** 1Department of Psychology, University of Milan-Bicocca, 20126 Milan, Italyfranca.crippa@unimib.it (F.C.);; 2Department of Biomedical Sciences for Health, Section of Legal Medicine and Insurance, University of Milan, 20122 Milan, Italy; 3Department of Adult Psychiatry, Nimes University Hospital, 30900 Nimes, France

**Keywords:** suicide, caustic ingestion, violent suicide, nonviolent suicide, suicide methods, Italy

## Abstract

Background: Suicide methods chosen by victims are particularly critical in suicide risk research. To differentiate suicide deaths, it is usual to categorize them as violent and nonviolent depending on the detrimental method chosen by the victims. Caustic ingestion, for example, is traditionally considered as a nonviolent suicide method. It results in severe consequences for the human body and it is associated with high levels of lethality. Methods: In this study, we retrospectively analyzed suicides that occurred between 1993 and 2021 in Milan (Italy) and that underwent autopsy. We compared a sample of 40 victims that ingested caustic substances with a sample of 460 victims of other chemical ingestion, and a sample of 3962 victims from violent suicide. Univariate analyses and univariate logistic regression models were performed. Suicides from caustic poisoning were significantly older, had a higher mean number of diseases and were more affected by psychiatric diseases compared to other chemical ingestion victims. By contrast, caustic suicides, compared to violent suicides, had a more balanced gender ratio, a higher mean number of diseases, were more affected by psychiatric diseases, had a higher rate of complex suicides (more than one modality), and had victims who died more frequently inside instead of outside. In logistic regression models, age was the only feature differentiating caustic from other chemical ingestion suicides while the features differentiating caustic from violent suicides were gender, mean number of diseases and suicide place. Conclusions: Suicides by caustic ingestion showed substantial differences compared to violent suicides, with a higher severe profile. However, some differences were reported comparing caustic ingestion to other chemical ingestion as well. Thus, we argue whether it is more appropriate to differentiate the suicidal ingestion of caustics from both violent and nonviolent suicide methods.

## 1. Introduction

Suicide is a leading public health concern causing the death of about 4000 people each year in Italy [1] and 700,000 people every year in the world [2]. The north of Italy was found to be the area of greatest concern due to higher suicide rates [1] and includes, among other cities, the wide area of Milan and its provinces.

Many efforts have been made in research to identify risk factors that drive people to suicide. Biology might play a key role determining a certain degree of familiarity for specific genes that predispose individuals to suicide [3,4]. Together with the heritability of suicide behaviors, psychosocial risk factors have a huge impact on the complexity of suicidality [5], highlighting the importance of the psychological vulnerability [6] as well as the relationship between personality traits and the neurophysiological functioning [7,8]. Moreover, socioeconomical and environmental problems can lead to additional psychological distress and exacerbate predispositions toward suicide [9]. To reduce the number of suicides, research has put its efforts on prevention by identifying different categories of suicidal behaviors and, therefore, the factors that may lead people to choose to use one method over another. The Åsberg classification [10] is the most widely categorization used in suicide research, which traditionally relies on the method used by the victims to distinguish between different subtypes of suicide. According to this classification, suicides can be clustered into violent and nonviolent: Violent methods include hangings, the use of firearms, jumps from heights, deep cuts, car accidents, burning, gas poisoning, drowning, electrocution, and jumping under a train; while among the nonviolent methods are gathered overdose and the use of toxic substances. Although the Åsberg [10] classification is quite dated, only a few authors have proposed modifications, for example by including in the nonviolent suicide methods such as poison by gases, suffocation [11], and drowning [12]. However, some aspects of the classification of violent and nonviolent clusters remain mixed. For instance, violent suicide methods condense some of the most painful or disfiguring modality of death and are related to an elevated suicide risk [13,14]. Furthermore, the consequences in both external and internal body injuries are different from violent and nonviolent methods and the risk of death by suicide differs according to the method [13]. In general, violent methods are characterized by a higher lethality compared to nonviolent methods [13,15]. Moreover, violent suicide victims appear to exhibit some demographic and clinical characteristics [16], such as older age [14,17] and higher rates of mental disorders [15]. The number of violent suicides increases with age and is more pronounced in men, and there seems to be some degree of seasonality in violent suicides but not in nonviolent suicides [18]. The male gender is a steadily reported major factor for violent suicide, who most often use modes such as shooting, hanging or drowning [19]. People who choose violent methods are also more prone to risky decision-making [20].

Ingestion of corrosive agents with the intention to die by suicide is not rare [21]: In Italy, 5.8% of the male population dies each year from toxic poisoning, while a significantly higher percentage of women, corresponding to 12.1%, kill themselves through toxic poisoning [1]. A proper categorization of methods of suicide is essential to investigate the factors that increase the risk of suicide and, thus, improve appropriate assessment, treatment, and health care policies for those at risk. Moreover, research that relies on these distinctions highly reflects how suicides are classified. 

Typically, the literature treats nonviolent methods as a homogeneous cluster that include, among others, suicide by substance poisoning such as pesticide ingestion, gas poisoning, caustic ingestion, and overdose [11,12,22]. However, among the toxic substances that could be ingested, some might be more dangerous than others. In particular, caustic agents, which are widely used in domestic, industrial and agricultural settings, could be responsible for short- and long-term complications and seem to show higher mortality, especially when associated with older age and higher comorbidity scores [23,24]. In individuals who die after caustic exposure, a spectrum of specific and predictable injuries is seen to the respiratory and gastrointestinal tracts [25], ranging from burns, massive hemorrhage, and perforation with aorto-enteric, gastrocolic, or gastro-bronchial fistulas resulting in permanent damages in case of survival [26]. However, despite the severity of the injuries, this method of suicide is yet categorized as a nonviolent method. Thus far, no research has examined the characteristics of caustic ingestion deaths. For this reason, this study had two aims:(1)We first wanted to investigate the differences in terms of socio-demographic, clinical and suicide-related features between caustic ingestion and other chemical ingestion. In research, caustic ingestion and chemical ingestion are commonly treated as the same suicide method, although the similarities and differences between these methods have not yet been explored in depth.(2)In the second analysis, we compared caustic ingestion victims to violent suicides to better understand the relationship between these two suicide methods. Similar to step (1), we analyzed socio-demographic, clinical and suicide-related features.

## 2. Materials and Methods

In this study, a retrospective analysis was carried out comparing three groups of suicide cases. Group 1 included 40 cases that occurred due to ingestion of caustics. To classify caustics, we followed Zoja’s classification of caustics [27]. See Table 1 for strong acids (pH < 2), such as muriatic and sulfuric; strong bases (pH > 12), for example ammonia or lye (caustic potash); and oxidizing agents, such as bleach (sodium hypochlorite). Group 2 consisted in 460 cases that died due to chemical exposure to other substances: ingestion of drugs (antidepressants, anxiolytics, benzodiazepines, sedatives), addictive substances (heroin and cocaine), other chemical agents (potassium chloride, herbicides, rat poison, brake oil as ethylene glycol or trichloroethylene), or inhalation of gas (nitrogen, methane, propane, butane, helium, carbon monoxide). Group 3 included 3962 cases that died through violent methods (e.g., hangings, the use of firearms, jumps from heights, car accidents, burning, jumping under a train). A flowchart of the enrolled subjects is reported in Figure 1. The database of the Institute of Forensic Medicine of Milan provided the data of the autopsies performed between 1993 and 2021. All the data came from the certified copies of the originals deposited with the prosecutors of the courts through the autopsy reports, annual registers and historical archive and autopsy reports released by the Institute of Forensic Medicine.

The groups were compared in terms of sociodemographic features, such as gender, age, and ethnicity; to simplify, ethnicity was generally grouped in white Caucasian vs. other ethnicities. The groups were also compared in terms of clinical features, occurrence of both psychiatric and organic disorder(s); comorbidities; medication assumption; and in terms of psychotropic medication, such as antidepressants, anxiolytics, benzodiazepines, sedatives, and non-psychotropic medication, such as chemotherapeutics, cardiotonics, antiretrovirals, antihypertensives, anticonvulsants, and hypoglycemics. Moreover, the groups were compared in terms of suicide-related features, suicidal ideation and/or attempts, suicide place, simple vs. complex suicide, causes of death (circulatory failure, intoxication, and organic injury) and suicide place, the latter clustered in inside (home, hospital, work, car/garage, prison) and outside (binaries, street, open space such as green area, park or field, river or lakes, and public area). Medico-legally speaking, suicidal ideation was defined as having verbally expressed to someone the intention to die by suicide; instead, previous suicide attempts were defined as having engaged in concrete acts of self-harm. Indeed, as it is usually shown in the literature, simple suicides and complex suicides have here been distinguished [28,29,30]: For simple suicide we meant the application of a single injurious modality while for complex suicide we referred to the use of more than one modality simultaneously or in chronological succession. Diseases were categorized according to the ICD-10 classification [31]. All information was collected post-mortem by medical professionals in charge of autopsies. 

The IBM SPSS statistics 27 software (IBM Corp., Armonk, NY, USA) was used for statistical analyses. The groups were compared with regard to the socio-demographic, clinical, and suicide-related variables. We tested the normal distribution of the variables. Since all the continuous variables were not normally distributed, we used the Mann-Whitney U test only, while Chi-square test was applied for categorical variables. Univariate logistic regression models were used as well. We included in the models (Group 1 vs. Group 2 and Group 1 vs. Group 3) the variables that were found to be different among the groups in the univariate analyses. The analysis was conducted by setting statistical significance at *p* ≤ 0.05.

The subjects involved in this study underwent a judicial autopsy at the Institute of Legal Medicine of Milan to identify the cause of death. Data collecting, sampling and subsequent forensic analysis were authorized by the public prosecutor. Therefore, data were acquired as part of a forensic judicial investigation and in accordance with Italian Police Mortuary Regulation. In accordance with Italian law, ethical approval is not required in these cases, however, the anonymity of the subjects must be guaranteed.

## 3. Results

### 3.1. Caustic vs. Chemical Suicides

The sample consisted of victims from caustic ingestion (Group 1, *n =* 40) and of victims from chemical ingestion of other substances (Group 2, *n=* 460).

#### 3.1.1. Socio-Demographic Features

The cases of Group 1 were older than Group 2 ones (*U* = −3.56, *p* < 0.001). However, Group 1 and Group 2 did not differ in terms of sex (χ^2^ (1) = 2.83, *p* = 0.09) and ethnicity (χ^2^ (1) = 0.11, *p* = 0.74). See Table 2 for details.

#### 3.1.2. Clinical Features

As shown in Table 3, the two groups differ in the mean number of diseases: The cases in Group 1 were reported to have had significantly more diseases on average than Group 2 (*U =* −2.14, *p* = 0.03). Group 1 was significantly more affected by psychiatric disorders (χ^2^ (1) = 4.0, *p* = 0.05) than Group 2. No further significant differences have been found between the two groups. Details about the organic diseases are presented in Table 4.

#### 3.1.3. Suicide-Related Features

Concerning suicide-related features, no significant difference was found between Group 1 and Group 2, referring to previous suicidal ideation (χ^2^ (1) = 0.04, *p* = 0.84), previous suicide attempt(s) (χ^2^ (1) = 0.23, *p* = 0.64), and the method of suicide (simple vs. complex; χ^2^ (1) = 0.22, *p* = 0.64; Table 5). See Table 6 for more details about the type of chemical ingested and the causes of death. 

#### 3.1.4. Univariate Logistic Regression

Explanatory variables showed no multicollinearity, all correlation values less than 0.5. Considering among covariants age, number of diseases, and presence of psychiatric disorder(s), the only variable significantly differentiating Group 1 from Group 2 in the univariate logistic regression model was age (OR: 1.03; 95% CI: 1.01–1.06; *p*: 0.002) (Table 8). Group 1 had a slightly higher mean age compared to Group 2. Goodness of fit estimates are indicated in Table 7.

### 3.2. Caustic vs. Violent Suicides

We compared victims of caustic ingestion (Group 1, *n =* 40) and victims of violent suicide (Group 3, *n =* 3962).

#### 3.2.1. Socio-Demographic Features

The two groups differ in sex, with Group 3 predominantly composed of men compared to Group 1 (χ^2^ (1) = 11.06, *p* = 0.001). However, Group 1 and Group 3 did not differ in terms of age (*U* = −1.83, *p* = 0.07) and ethnicity (χ^2^ (1) = 0.54, *p* = 0.46). See Table 2 for details.

#### 3.2.2. Clinical Features

Statistically significant differences between Group 1 and Group 3 were found on the mean number of diseases (Table 3), where Group 1 reported to have had more diseases on average than Group 3 (*U =* −2.63, *p* < 0.01). Moreover, Group 1 shown to be significantly more affected by psychiatric disorder(s) than Group 3 (χ^2^ (1) = 5.5, *p* < 0.02). Details about the organic diseases are presented in Table 4. 

#### 3.2.3. Suicide-Related Features

As shown in Table 5, significant differences were found between Group 1 and Group 3 for complex suicide, where Group 1 showed a higher number of complex suicides (χ^2^ (1) = 5.27, *p* < 0.02). Furthermore, concerning the suicide place, the number of suicides inside were significantly higher in Group 1 victims rather than in Group 3 victims (χ^2^ (1) = 20.09, *p* < 0.001). See Table 8 for details. No further differences have been found between the two groups. 

#### 3.2.4. Univariate Logistic Regression

Explanatory variables showed no multicollinearity, all correlation values less than 0.5. Considering among covariants sex, number of diseases, presence of psychiatric disorder(s), presence of complex suicide, and suicide place (inside vs. outside), the variables significantly differentiating Group 1 from Group 3 in the univariate logistic regression model were sex (OR: 3.21; 95% CI: 1.68–6.14; *p* < 0.001), mean number of diseases (OR: 1.53; 95% CI: 1.09–2.15; *p*: 0.01) and suicide place (OR: 8.03; 95% CI: 3.10–20.79; *p* < 0.001) (Table 8). ORs showed that there were high differences between the two groups, in particular Group 1 suicides took place mainly indoor, and Group 1 subjects were more often women and reported a higher number of diseases. Goodness of fit estimates are indicated in Table 7.

## 4. Discussion

The aim of this study was to compare victims of suicide by caustic ingestion to victims of suicide by other chemical ingestion and to victims of violent suicide in order to evaluate the most appropriate categorization of caustic ingestion as a method of suicide. Caustic ingestion is traditionally classified as a nonviolent method of suicide [10], thus our study revealed some demographic, clinical and suicidal features that differentiate these victims from nonviolent victims as well as violent suicide deaths. 

Previous studies have shown the enhancement of suicide risk among the elderly [32,33], which increases as the years go by. We found that victims that ingested caustics were significantly older compared to cases that died from the ingestion of other chemical substances, highlighting an increased risk of completing suicide for these individuals. Moreover, results showed that those who died from caustic ingestion had a higher mean number of diseases and were significantly more affected by psychiatric disease(s). As previously reported, these could include depression, schizophrenia and personality disorders [34,35]. Comorbidity considerably compounds the clinical picture, and it could influence the choice of a potentially more harmful and lethal method of suicide. Certain differences were also observed from the comparison between caustic ingestion victims and violent suicides. For example, caustic ingestion victims had a more balanced gender ratio while violent suicide victims were most frequently males. This result is in line with previous studies showing that men are more likely to choose a violent method of suicide [36].

Several studies have investigated the relationship between physical illness and suicide risk [37,38,39]. According to our results, the mean number of diseases was higher for caustic ingestion victims. The caustic ingestion sample was significantly more affected by illnesses compared to violent suicides, a relevant finding showing that this sample may be exposed to higher risks of suicide. Moreover, complex suicide is often associated with fatal outcomes [40] and appeared to be more frequent for caustic ingestion victims, highlighting a much more serious clinical picture of caustic suicides than the violent sample. Nevertheless, this result was obtained from a very limited sample of cases (2 cases in Group 1 compared to 46 cases in Group 3) and, for this reason, it should be considered with caution. Another distinctive feature of suicides through caustics concerned the place of suicides, in fact, suicides inside were more frequent for caustic victims compared to the violent suicides sample. Dying indoors should not be underestimated, as it could be a distinguishing feature of the most at-risk individuals.

According to regression analyses, we found no substantial differences between caustic and chemical suicides, the two groups appear not to significantly differ except for age. In fact, similarly to violent suicide [15,16], our caustic sample appeared to be older compared to chemical poisoning victims. As mentioned earlier, age has been frequently linked to suicide [32,33], but this was the single feature that distinguished the two samples.

However, regression analyses of caustic vs. violent suicides showed more relevant differences. For instance, caustic ingestion victims had a more balanced gender ratio. Previous studies frequently associate the male gender with violent suicides and a high fatality of the suicide method [13,14,41], but yet recent studies found a strong association with self-poisoning and in particular with caustics in men as well [42]. According to our finding, women may have the same risk of ingesting caustics than men, therefore this group of individuals may be at greater risk of ingesting fatal poisons compared to other methods.

Results point out that suicides by caustics seem to differ from violent suicides, but might also show a more severe clinical picture: Our regression analysis highlights the higher mean number of diseases in suicides by caustics, corroborating the idea that these victims may be exposed to more consistent risk factors. Previous studies already showed that physical illness and functional impairment are relevant elements that increase the risk of suicide [38,39,43]. In addition, from our regression analysis, caustic suicide victims appear to choose more frequently places, such as home, hospitals, cars, garages, prisons, hotels, and work places to kill themselves compared to violent suicide victims. The suicide method has a major impact on the likelihood of a fatal outcome: This is a factor of great concern in suicide prevention since specific characteristics of victims have been observed to be associated with the choice of particular methods of suicide [44]. However, choosing to die in an isolated place may increase the likelihood of a fatal outcome, as the chances of being rescued decrease. The factors that lead to suicide are diverse and multifaceted and it is critical that suicides are properly clustered to aid the implementation of more accurate prevention and intervention strategies. The literature [10,11,12,22] describes violent methods as all those methods that lead to extremely dangerous consequences to the body, such as jumping from heights, deep cuts, car accidents and jumping under a train, burning, electrocution, and gunshot wounds. By contrast, nonviolent suicides are often summarized as poisoning, overdose or toxic substances ingestion without considering the differences between the many toxic substances that can be ingested. Among these, caustics in particular differ from other toxicants due to their high destructive power, being able to cause burns, visceral perforations and hemorrhages associated with severe pain. Caustic injuries are also burdened with a high complication rate and characterized by particularly high mortality. This is also a consequence of the fact that there are no antidotes for caustics and, as liquids, they flow very quickly through the digestive tract and spread throughout over the entire transit area [27].

These considerations corroborate the idea that caustics ingestion as a means of suicide differ from other chemical ingestion nonviolent suicides. Therefore, we suggest that suicides by caustic ingestion should be categorized differently, and perhaps it would be more appropriate to consider them separately from both violent and nonviolent suicide victims following other chemical ingestion. 

In this study we aimed to search for a more defined and accurate categorization of suicide methods taking into consideration factors which have previously been underestimated, such as the choice of the different substances used to die by suicide. We propose that the vast differences in the effects on the human body from the ingestion of one toxic substance to another are a relevant factor in the subject’s choice of suicide method. The ingestion of caustic substances has one of the highest death rates [23,24], and the victims that use this method differ not only from violent suicides but also from victims of nonviolent suicides (e.g., victims of chemical ingestion). 

## 5. Conclusions

To the best of our knowledge this is the first study comparing suicide deaths that occurred due to caustic ingestion vs. other chemical ingestion vs. violent methods. Also considering the severity of the injuries caused by caustics, we conclude by proposing that caustic ingestion might be better addressed by differentiating it from both violent and nonviolent methods of suicide.

### Limitations

This study has several limitations. We acknowledge that the number of victims of caustics poisoning analyzed was limited and this may have impacted our results. Future studies might consider a larger sample of victims of caustics ingestion to report more consistent findings. Moreover, all the data were collected post-mortem, therefore relevant clinical or anamnestic information may have been incomplete. 

## Figures and Tables

**Figure 1 ijerph-20-06270-f001:**
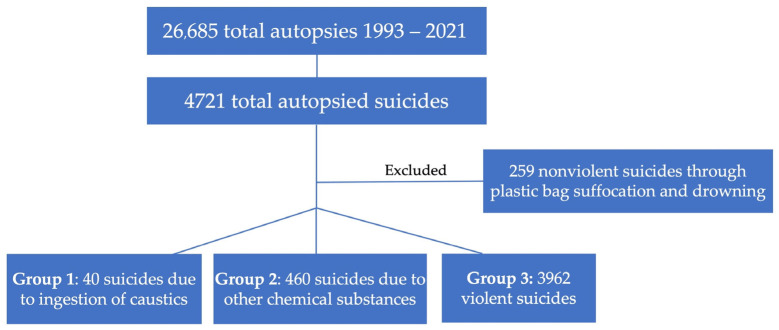
Flowchart of the enrolled subjects.

**Table 1 ijerph-20-06270-t001:** Types of caustic agents according to Zoja’s classification [27].

Category	Chemical	Common Name	Use
Strong acids (pH < 2)	sulfuric acid	vitriol	detergent; in batteries
	hydrochloric acid	muriatic	detergent
	nitric acid	etching	detergent
	phosphoric acid		detergent
	oxalic acid		anti-rust
Strong bases (pH > 12)	sodium hydroxide	caustic soda	detergent
	potassium hydroxide	caustic potash	detergent
	ammonium hydroxide	ammonia	detergent
Oxidizing agents	sodium hypochlorite	bleach	whitener
	phenols		
	tincture of iodine		
	hydrogen peroxide	hydrogen peroxide	whitener
	potassium permanganate		

**Table 2 ijerph-20-06270-t002:** Sociodemographic data of the samples in Group 1, Group 2, Group 3, the total sample and statistics.

Variables	Group 1 (*n =* 40) *n* (%) or Mean ± SD	Group 2 (*n =* 460) *n* (%) or Mean ± SD	Group 1–2 Statistics Chi-2 (df), *p* or Mann-Whitney U Test, *p*	Group 3 (*n =* 3962) *n* (%) or Mean ± SD	Group 1–3 Statistics Chi-2 (df), *p* or Mann-Whitney U Test, *p*	Total Sample (*n* = 4462) *n* (%) or Mean ± SD
SEX			2.83 (1), 0.09		11.06 (1), 0.001	
Men	19 (47.5%)	281 (61.1%)		2830 (71.4%)		3130 (70.1%)
Women	21 (52.5%)	179 (38.9%)		1132 (28.6)		1332 (29.9%)
AGE	56.82 ± 16.36	47.43 ± 16.16	−3.56, <0.001	51.73 ± 19.31	−1.83, 0.07	51.33 ± 19.03
ETHNICITY			0.11 (1), 0.74		0.54 (1), 0.46	
WHITE CAUCASIAN	39 (97.5%)	444 (96.5%)		3731 (94.9%)		4214 (94.4%)
Albanian	-	-		17 (0.4%)		17 (0.4%)
Austrian	-	-		2 (0.05%)		2 (0.05%)
American	-	1 (0.2%)		3 (0.08%)		4 (0.09%)
Bosnian	-	-		2 (0.05%)		2 (0.05%)
Bulgarian	-	-		5 (0.1%)		5 (0.1%)
Canadian	-	-		1 (0.03%)		1 (0.02%)
Czech	-	-		2 (0.05%)		2 (0.05%)
Croatian	-	-		3 (0.08%)		3 (0.07%)
Danish	-	1 (0.2%)		-		1 (0.02%)
English	-	4 (0.9%)		17 (0.4%)		21 (0.05%)
Estonian	-	-		2 (0.05%)		2 (0.05%)
Finnish	-	1 (0.2%)		1 (0.03%)		2 (0.05%)
French	-	-		8 (0.2%)		8 (0.02%)
Georgian	-	-		1 (0.03%)		1 (0.02%)
German	1 (2.5%)	4 (0.9%)		15 (0.4%)		20 (0.05%)
Greek	-	1 (0.2%)		3 (0.08%)		4 (0.09%)
Hungarian	-	1 (0.2%)		2 (0.05%)		3 (0.07%)
Italian	37 (92.5)	419 (91.1%)		3549 (89.6%)		4005 (89.8%)
Latvian	1 (2.5%)	-		2 (0.05%)		3 (0.07%)
Maltese	-	-		2 (0.05%)		2 (0.05%)
Norwegian	-	2 (0.4%)		1 (0.03%)		3 (0.07%)
Dutch	-	-		3 (0.1%)		3 (0.07%)
Polish	-	3 (0.7%)		8 (0.2%)		11 (0.2%)
Poland	-	-		1 (0.03%)		1 (0.02%)
Portuguese	-	-		3 (0.08%)		3 (0.07%)
Romanian	-	2 (0.4%)		33 (0.8%)		35 (0.8%)
Russian	-	1 (0.2%)		3 (0.08%)		4 (0.09%)
Slovak	-	2 (0.4%)		-		2 (0.05%)
Slovenian	-	-		3 (0.08%)		3 (0.07%)
Somali	-	-		1 (0.03%)		1 (0.02%)
Spanish	-	1 (0.2%)		14 (0.4%)		15 (0.3%)
Swedish	-	-		2 (0.05%)		2 (0.05%)
Swiss	-	-		3 (0.08%)		3 (0.07%)
Ukrainian	-	1 (0.2%)		19 (0.5%)		20 (0.5%)
OTHER ETHNICITIES	1 (2.5%)	16 (3.5%)		199 (5.1%)		216 (4.8%)
Afghan	-	-		2 (0.05%)		2 (0.05%)
Algerian	-	1 (0.2%)		5 (0.1%)		6 (0.1%)
Angolan	-	-		1 (0.03%)		1 (0.02%)
Arab	1 (2.5%)	1 (0.2%)		11 (0.3%)		13 (0.3%)
Argentinian	-	1 (0.2%)		2 (0.05%)		3 (0.07%)
Armenian	-	-		1 (0.03%)		1 (0.02%)
Bangladeshi	-	-		4 (0.1%)		4 (0.09%)
Bolivian	-	-		3 (0.08%)		3 (0.07%)
Brazilian	-	1 (0.2%)		8 (0.2%)		9 (0.2%)
Burundian	-	-		1 (0.03%)		1 (0.02%)
Chilean	-	-		4 (0.1%)		4 (0.09%)
Chinese	-	1 (0.2%)		12 (0.3%)		13 (0.3%)
Colombian	-	-		3 (0.08%)		3 (0.07%)
Korean	-	-		4 (0.1%)		4 (0.09%)
Cuban	-	-		6 (0.2%)		6 (0.1%)
Ecuadorian	-	1 (0.2%)		4 (0.1%)		5 (0.1%)
Egyptian	-	-		15 (0.4%)		15 (0.03%)
Eritrean	-	-		5 (0.1%)		5 (0.1%)
Ethiopian	-	-		4 (0.1%)		4 (0.09%)
Philippine	-	-		15 (0.4%)		15 (0.03%)
Gambles	-	-		1 (0.03%)		1 (0.02%)
Ghanaian	-	-		1 (0.03%)		1 (0.02%)
Japanese	-	-		2 (0.05%)		2 (0.05%)
Guinean	-	-		1 (0.03%)		1 (0.02%)
Haitian	-	-		1 (0.03%)		1 (0.02%)
Indian	-	1 (0.2%)		5 (0.1%)		6 (0.1%)
Indonesian	-	-		1 (0.03%)		1 (0.02%)
Iranian	-	-		3 (0.08%)		3 (0.07%)
Ivorian	-	-		1 (0.03%)		1 (0.02%)
Kazaks	-	1 (0.2%)		-		1 (0.02%)
Kenyan	-	-		1 (0.03%)		1 (0.02%)
Malian	-	-		4 (0.1%)		4 (0.09%)
Moroccan	-	3 (0.7%)		15 (0.4%)		18 (0.04%)
Moldavian	-	-		5 (0.1%)		5 (0.1%)
Mozambican	-	-		1 (0.03%)		1 (0.02%)
Nepalese	-	-		1 (0.03%)		1 (0.02%)
Nigerian	-	2 (0.4%)		3 (0.08%)		5 (0.1%)
Pakistani	-	-		2 (0.05%)		2 (0.05%)
Paraguayan	-	-		1 (0.03%)		1 (0.02%)
Peruvian	-	3 (0.7%)		10 (0.3%)		13 (0.3%)
Salvadoran	-	-		5 (0.1%)		5 (0.1%)
Senegalese	-	-		3 (0.08%)		3 (0.07%)
Sinhalese	-	-		9 (0.2%)		9 (0.2%)
Tunisian	-	-		3 (0.08%)		3 (0.07%)
Turkish	-	-		7 (0.2%)		7 (0.02%)
Venezuelan	-	-		1 (0.03%)		1 (0.02%)
Zambezian	-	-		2 (0.05%)		2 (0.05%)

**Table 3 ijerph-20-06270-t003:** Details of the clinical features in Group 1, Group 2, Group 3, the total sample, and statistics.

Variables	Group 1 (*n =* 40) *n* (%) or Mean ± SD	Group 2 (*n =* 460) *n* (%) or Mean ± SD	Group 1–2 Statistics Chi-2 (df), *p* or Mann-Whitney U test, *p*	Group 3 (*n =* 3962) *n* (%) or Mean ± SD	Group 1–3 Statistics Chi-2 (df), *p* or Mann-Whitney U test, *p*	Total Sample (*n* = 4462) *n* (%) or Mean ± SD
MEDICAL HISTORY	36 (90%)	366 (79.6%)	2.54 (1), 0.11	3098 (78.2%)	3.25 (1), 0.07	3500 (78.4%)
SINGLE DISEASE	20 (55.6%)	242 (66.1%)	1.61 (1), 0.2	2103 (67.9%)	2.48 (1), 0.12	2365 (53%)
MULTIPLE DISEASE	16 (44.4%)	124 (33.9%)		995 (32.1%)		1135 (25.4%)
MEAN NUMBER OF DISEASES	1.58 ± 1.2	1.18 ± 0.9	−2.14, 0.03	1.11 ± 0.85	−2.63, 0.009	1.12 ± 0.86
PSYCHIATRIC DISORDER	33 (82.5%) ^a^	309 (67.2%) ^a^	4.0 (1), 0.05 ^b^	2564 (64.7%) ^a^	5.5 (1), 0.02 ^b^	2906 (65.1%) ^a^
Depression	28 (70%)	249 (54.1%)		2114 (53.4%)		2391 (53.5%)
Bipolar Disorder	1 (2.5%)	9 (2%)		88 (2.2%)		98 (2.2%)
Psychotic Disorder	5 (12.5%)	36 (7.8%)		267 (6.7%)		308 (6.9%)
Anorexia Nervosa	-	6 (1.3%)		23 (0.6%)		29 (0.6%)
Personality Disorder	-	4 (0.9%)		-		4 (0.1%)
Anxiety Disorder	-	4 (0.9%)		91 (2.3%)		95 (2.1%)
Alcoholism	2 (5%)	24 (5.2%)		163 (4.1%)		189 (4.2%)
Addiction	-	34 (7.4%)		195 (4.9%)		229 (5.1%)
Organic mental disorders (Autism, Mental retardation, Dementia)	-	1 (0.2%)		8 (0.2%)		9 (0.2%)
Multiple psychiatric disorders	3 (7.5%)	47 (10.2%)		339 (8.6%)		389 (18.7%)
MEAN NUMBER OF PSYCHIATRIC DISORDERS	0.9 ± 0.5	0.8 ± 0.68	−1.41, 0.16	0.7 ± 0.64	−1.91, 0.06	0.8 ± 0.64
ORGANIC DISEASE	16 (40%) ^a^	129 (28%) ^a^	2.56 (1), 0.11 ^b^	1089 (27.5%) ^a^	3.1 (1), 0.08 ^b^	1234 (27.7%) ^a^
Infectious and parasitic diseases	1 (2.5%)	23 (5%)		106 (2.7%)		130 (2.9%)
Neoplasms	3 (7.5%)	26 (5.7%)		285 (7.2%)		314 (7%)
Endocrine, nutritional and metabolic diseases	2 (5%)	28 (6.1)		168 (4.2)		198 (4.4%)
Diseases of the nervous system	3 (7.5%)	17 (3.7%)		111 (2.8)		131 (2.9%)
Diseases of the eye and adnexa	1 (2.5%)	2 (0.4%)		28 (0.7%)		31 (0.7%)
Diseases of the ear and mastoid process	-	1 (0.2%)		4 (0.1%)		5 (0.1%)
Diseases of the circulatory system	6 (15%)	40 (8.7%)		428 (10.8%)		474 (10.6%)
Diseases of the respiratory system	1 (2.5%)	10 (2.2%)		85 (2.1%)		96 (2.2%)
Diseases of the digestive system	1 (2.5%)	13 (2.8%)		60 (1.5%)		74 (1.7%)
Diseases of the skin and subcutaneous tissue	-	-		4 (0.1)		4 (0.1%)
Diseases of the musculoskeletal system and connective tissue	1 (2.5%)	5 (1.1%)		42 (1.1%)		48 (1.1%)
Diseases of the genitourinary system	1 (2.5%)	-		23 (0.6%)		24 (0.5%)
Congenital malformations, deformations and chromosomal abnormalities	1 (2.5%)	-		3 (0.1%)		4 (0.1%)
Factors influencing health status and contact with health services	-	7 (1.5%)		32 (0.8%)		39 (0.9%)
Multiple organic disorders	5 (12.5%)	36 (90%)		276 (8.2%)		317 (7.1%)
MEAN NUMBER OF ORGANIC DISEASES	0.68 ± 1.12	0.39 ± 0.7	−1.67, 0.1	0.37 ± 0.67	−1.85, 0.06	0.4 ± 0.7
ANY MEDICATION	24 (60%)	264 (57.4%)	0.10 (1), 0.75 ^c^	2108 (53.2%)	0.73 (1), 0.39	2396 (53.7%)
PSYCHOTROPIC MEDICATION	21 (52.5%)	235 (51.1%)	0.03 (1), 0.86 ^c^	1726 (43.6%)	1.29 (1), 0.26	1982 (44.4%)
OTHER MEDICATION	5 (12.5%)	44 (9.6%)	0.36 (1), 0.55 ^c^	472 (11.9%)	0.01 (1), 0.91	521 (11.7%)

^a^ Since several subjects had multiple diseases, the column total number and percentages give more than the reported total numbers. ^b^ Differences between the two groups were analyzed by comparing those who had the disorder/disease with those who did not have the disorder/disease. ^c^ Differences between the two groups were analyzed by comparing those who take a medication with those who did not take a medication.

**Table 4 ijerph-20-06270-t004:** Details by anatomical district of the organic diseases in Group 1, Group 2, Group 3, and the total sample.

Variables	Group 1 (*n =* 40) *n* (%) or Mean ± SD	Group 2 (*n =* 460) *n* (%) or Mean ± SD	Group 3 (*n =* 3962) *n* (%) or Mean ± SD	Total Sample (*n* = 4462) *n* (%) or Mean ± SD
ORGANIC DISEASE	16 (40%) *	129 (28%) *	1089 (27.5%) *	1234 (27.7%) *
Infectious and parasitic diseases				
HIV+	-	12 (2.6%)	47 (1.2%)	59 (1.3%)
AIDS	-	1 (0.2%)	8 (0.2%)	9 (0.2%)
Poliomyelitis	-	2 (0.4%)	4 (0.1%)	6 (0.1%)
HBV	1 (2.5%)	-	8 (0.2%)	9 (0.2%)
HCV	-	11 (2.4%)	50 (1.3%)	61 (1.4%)
Neoplasms				
Tumor cachexia	-	1 (0.2%)	1 (0.03%)	2 (0.05%)
Leukemia	-	1 (0.2%)	6 (0.2%)	7 (0.2%)
Neoplasms of skin	-	-	2 (0.1%)	2 (0.05%)
Neoplasms of eye, brain and other parts of central nervous system	-	2 (0.4%)	15 (0.4%)	17 (0.4%)
Neoplasms of female genital organs	2 (5%)	-	19 (0.5%)	21 (0.5%)
Neoplasms of male genital organs	-	4 (0.9%)	37 (0.9%)	41 (0.9%)
Neoplasms of digestive organs	1 (2.5%)	6 (1.3%)	109 (2.8%)	116 (2.6%)
Neoplasms of respiratory and intrathoracic organs	-	1 (0.2%)	39 (1%)	40 (0.9%)
Neoplasms of lip, oral cavity and pharynx	-	-	23 (0.6%)	23 (0.5%)
Neoplasms of breast	-	5 (1.1%)	24 (0.6%)	29 (0.6%)
Neoplasms of the bone	-	3 (0.7%)	4 (0.1%)	7 (0.2%)
Neoplasms of thyroid and other endocrine glands	-	-	6 (0.2%)	6 (0.1%)
Endocrine, nutritional, and metabolic diseases				
Hyperlipidemia/ Dyslipidemia	-	-	4 (0.1%)	4 (0.1%)
Thyroid dysfunction	-	1 (0.2%)	16 (0.4%)	17 (0.4%)
Deficit alfa-1	-	-	1 (0.03%)	1 (0.02%)
Obesity	1 (2.5%)	2 (0.4%)	8 (0.2)	11 (0.2%)
Goiter	-	1 (0.2%)	-	1 (0.02%)
Diabetes	1 (2.5%)	22 (4.8%)	128 (3.2%)	151 (3.4%)
Hypercholesterolemia		1 (0.2%)	8 (0.2%)	9(0.2%)
Cystic fibrosis	-	1 (0.2%)	4 (0.1%)	5 (0.1%)
Diseases of the nervous system				
Epilepsy	2 (5%)	12 (2.6%)	44 (1.1%)	58 (1.3%)
Hemiparesis	1 (2.5%)	-	4 (0.1%)	5 (0.1%)
SLA	-	2 (0.4%)	7 (0.2%)	9 (0.2%)
Alzheimer’s disease	-	1 (0.2%)	15 (0.4%)	16 (0.4%)
Peripheral neuropathy	-	1 (0.2%)	-	1 (0.02%)
Myelitis	-	1 (0.2%)	-	1 (0.02%)
Neuropathy of the lower limbs	1 (2.5%)	1 (0.2%)	2 (0.1%)	4 (0.1%)
Headache	-	-	6 (0.1%)	6 (0.1%)
Corea di Huntington	-	-	3 (0.1%)	3 (0.1%)
Paralysis	-	-	8 (0.2%)	8 (0.2%)
Parkinson	-	-	24 (0.6%)	24 (0.5%)
Diseases of the eye and adnexa				
Blindness	-	-	7 (0.2%)	7 (0.2%)
Glaucoma	1 (2.5%)	1 (0.2%)	8 (0.2%)	10 (0.2%)
Maculopathy	-	1 (0.2%)	6 (0.2%)	7 (0.2%)
Coronophaty/Retinopathy	-	-	9 (0.2%)	9 (0.2%)
Diseases of the ear and mastoid process				
Tympanic perforation	-	1 (0.2%)	-	1 (0.02%)
Deaf-mutism	-	-	4 (0.1%)	4 (0.1%)
Diseases of the circulatory system				
Aneurysm	-	-	6 (0.2%)	6 (0.1%)
Hypertension/ Hypotension	3 (7.5%)	-	209 (5.3%)	212 (4.8%)
Cardiomyopathy	1 (2.5%)	18 (3.9%)	126 (3.2%)	145 (3.2%)
Cardiac Fibrillation	1 (2.5%)	1 (0.2%)	9 (0.2%)	11 (0.2%)
Myocardial infarction	1 (2.5%)	1 (0.2%)	36 (0.9%)	38 (0.9%)
Vasculopathy	-	1 (0.2%)	8 (0.2%)	9 (0.2%)
Atherosclerosis	1 (2.5%)	1 (0.2%)	5 (0.1%)	7 (0.2%)
Arrhythmia	-	2 (0.4%)	21 (0.5%)	23 (0.5%)
IPA	1 (2.5%)	15 (3.3%)	3 (0.1%)	19 (0.4%)
Stroke	1 (2.5%)	1 (0.2%)	39 (1%)	41 (0.9%)
Valvulopathy	-	1 (0.2%)	9 (0.2%)	10 (0.2%)
Varicocele	-	-	5 (0.1%)	5 (0.1%)
Diseases of the respiratory system				
Bronchial asthma	1 (2.5%)	2 (0.4%)	35 (0.9%)	38 (0.9%)
COPD - Chronic Obstructive Pulmonary Disease	-	6 (1.3%)	25 (0.6%)	31 (0.7%)
Pneumonia	-	1 (0.2%)	4 (0.1%)	5 (0.1%)
Bronchitis	-	1 (0.2%)	7 (0.2%)	8 (0.2%)
Emphysema	-	-	17 (0.4%)	17 (0.4%)
Pharyngitis	-	-	1 (0.03%)	1 (0.02%)
Diseases of the digestive system				
Gastric ulcer	1 (2.5%)	3 (0.7%)	25 (0.6%)	29 (0.6%)
Diverticulitis	-	1 (0.2%)	4 (0.1%)	5 (0.1%)
Gastritis	-	2 (0.4%)	5 (0.1%)	7 (0.2%)
Inguinal hernia	-	2 (0.4%)	4 (0.1%)	6 (0.1)
Rectal prolapse	-	1 (0.2%)	-	1 (0.02%)
Gastric band	-	2 (0.4%)	1 (0.03%)	3 (0.1%)
Cirrhosis	-	3 (0.7%)	10 (0.3%)	13 (0.3%)
Appendicitis	-	-	9 (0.2%)	9 (0.2%)
Celiac disease	-	-	2 (0.1%)	2 (0.05%)
Disease of the skin				
Psoriasis	-	-	4 (0.1%)	4 (0.1%)
Diseases of the musculoskeletal system and connective tissue				
Paget’s disease	1 (2.5%)	1 (0.2%)	-	2 (0.05%)
Rheumatoid arthritis	1 (2.5%)	1 (0.2%)	5 (0.1%)	7 (0.2%)
Osteoporosis	-	1 (0.2%)	10 (0.3%)	11 (0.2%)
Vertebral Fracture	-	1 (0.2%)	1 (0.03%)	2 (0.05%)
Hip prosthesis	-	1 (0.2%)	7 (0.2%)	8 (0.2%)
Arthritis	1 (2.5%)	2 (0.4%)	12 (0.3%)	15 (0.3%)
Disc herniation	-	1 (0.2%)	17 (0.4%)	18 (0.4%)
Diseases of the genitourinary system				
Prostatic hypertrophy	1 (2.5%)	-	17 (0.4%)	18 (0.4%)
Renal failure	-	-	6 (0.2%)	6 (0.1%)
Congenital malformations, deformations, and chromosomal abnormalities				
Turner Syndrome	1 (2.5%)	-	-	1 (0.02%)
Down Syndrome	-	-	1 (0.03%)	1 (0.02%)
Cystic Ovary Syndrome	-	-	2 (0.1%)	2 (0.05%)
Factors influencing health status and contact with health services				
Covid-19	-	4 (0.9%)	15 (0.4%)	19 (0.4%)
Hysterectomy	-	2 (0.4%)	-	2 (0.05%)
By-pass	-	1 (0.2%)	10 (0.3%)	11 (0.2%)
Homeless	-	-	7 (0.2%)	7 (0.2%)

* Since several subjects showed multiple diseases, the amount of total and percentage does not match the exact sum.

**Table 5 ijerph-20-06270-t005:** Suicide-related features in Group 1, Group 2, Group 3, the total sample, and statistics.

Variables	Group 1 (*n =* 40) *n* (%) or Mean ± SD	Group 2 (*n =* 460) *n* (%) or Mean ± SD	Group 1–2 Statistics Chi-2 (df), *p* or Mann-Whitney U Test, *p*	Group 3 (*n =* 3962) *n* (%) or Mean ± SD	Group 1–3 Statistics Chi-2 (df), *p* or Mann-Whitney U Test, *p*	Total Sample (*n* = 4462) *n* (%) or Mean ± SD
PREVIOUS SUICIDAL IDEATION	15 (37.5%)	180 (39.1%)	0.04 (1), 0.84 *	1191 (30.1%)	1.04 (1), 0.31 *	1386 (31%)
PREVIOUS SUICIDE ATTEMPT(S)	12 (30%)	155 (33.7%)	0.23 (1), 0.64 *	862 (21.8%)	1.58 (1), 0.21 *	1029 (23.1%)
SIMPLE SUICIDE	38 (95%)	428 (93%)	0.22 (1), 0.64	3918 (98.9%)	5.27 (1), 0.02	3984 (89.3%)
COMPLEX SUICIDE	2 (5%)	32 (7%)		46 (1.1%)		80 (1.8%)

* Differences between the two groups were analyzed by comparing those who reported previous suicidal ideation/suicide attempt(s) with those who did not report previous suicidal ideation/suicide attempt(s).

**Table 6 ijerph-20-06270-t006:** Details of the chemical ingested and the causes of death in Group 1 and Group 2.

Variables	Group 1 (*n =* 40) *n* (%) or Mean ± SD	Group 2 (*n =* 460) *n* (%) or Mean ± SD	Total Sample (*n* = 500) *n* (%) or Mean ± SD
Type of chemical ingested			
Strong acids	32 (80%)	-	32 (0.7%)
Strong bases	4 (10%)	-	4 (0.08%)
Oxidizing agents	1 (2.5%)	-	1 (0.02%)
Mix of different caustics	1 (2.5%)	-	1 (0.02%)
Medication	-	260 (56.5%)	260 (5.8%)
Gas	-	137 (29.8%)	137 (3.1%)
Drugs	-	8 (1.7%)	8 (0.2%)
Other chemical agents	-	6 (1.3%)	6 (0.1%)
Mix of different chemicals		17 (3.7%)	17 (3.4%)
Causes of death			
CIRCULATORY FAILURE	9 (22.5%)	289 (62.8%)	298 (59.6%)
Acute circulatory insufficiency	9 (22.5%)	273 (59.3%)	282 (56.4%)
Acute cardio-respiratory insufficiency	-	16 (3.5%)	16 (3.2%)
INTOXICATION	11 (27.5%)	149 (32.4%)	160 (32%)
Acute caustic intoxication	11 (27.5%)	-	11 (2.2%)
Gas poisoning	-	143 (31.1%)	143 (28.6%)
Substance poisoning	-	6 (1.3%)	6 (1.2%)
ORGANIC INJURY	20 (50%)	21 (4.7%)	41 (8.2%)
Visceral injuries from caustic ingestion	14 (35%)	-	14 (2.8%)
Multiple skeletal and visceral injuries	1 (2.5%)	3 (0.7%)	4 (0.8%)
Peritonitis	1 (2.5%)	3 (0.7%)	4(0.8%)
Septic state	4 (10%)	-	4(0.8%)
Edema	-	5 (1.1%)	5(1%)
Drowning	-	2 (0.4%)	2(0.4%)
Suffocation	-	4 (0.9%)	4(0.8%)
Hanging	-	4 (0.9%)	4(0.8%)

**Table 7 ijerph-20-06270-t007:** Univariate logistic regression analyses: comparison between Group 1 and Group 2 and Group 1 and Group 3.

Variables	B	S.E.	Wald	*p*	Exp(B)	95% C.I. for Exp(B)	
						Lower	Upper
Group 1 vs. Group 2							
Age	0.033	0.011	9.509	0.002	1.033	1.012	1.055
Number of diseases	0.151	0.173	0.763	0.382	1.163	0.828	1.634
Psychiatric disorder	0.678	0.462	2.157	0.142	1.970	0.797	4.867
Constant	−4.860	0.707	47.236	<0.001	0.008		
Goodness of fit: chi-square (3) = 16.64, *p* < 0.001, loglikelihood = 262.13							
Group 1 vs. Group 3							
Gender	1.168	0.331	12.470	<0.001	3.214	1.681	6.145
Number of diseases	0.427	0.172	6.149	0.013	1.532	1.094	2.147
Psychiatric disorder	0.415	0.448	0.857	0.355	1.515	0.629	3.648
Inside/outside	2.084	0.485	18.446	<0.001	8.033	3.104	20.788
Constant	−7.448	0.615	146.842	<0.001	0.001		
Goodness of fit: chi-square (4) = 46.64, *p* < 0.001, loglikelihood = 392.18							

**Table 8 ijerph-20-06270-t008:** Details of the suicide place for Group 1, Group 2, Group 3, the total sample, and statistics.

Variables	Group 1 (*n =* 40) *n* (%) or Mean ± SD	Group 2 (*n =* 460) *n* (%) or Mean ± SD	Group 1–2 Statistics Chi-2 (df), *p* or Mann-Whitney U Test, *p*	Group 3 (*n =* 3962) *n* (%) or Mean ± SD	Group 1–3 Statistics Chi-2 (df), *p* or Mann-Whitney U Test, *p*	Total Sample (*n* = 4462) *n* (%) or Mean ± SD
INSIDE	34 (87.2%)	429 (93.3%)	1.99 (1), 0.16	2026 (51.1%)	20.09 (1), <0.001	2.489 (55.8%)
Home	26 (65%)	259 (56.3%)		1527 (38.5%)		1812 (40.6%)
Hospital	8 (20%)	8 (1.7%)		182 (4.6%)		198 (4.4%)
Work	-	2 (0.4%)		136 (3.4%)		138 (3.1%)
Car/Garage	-	136 (29.6%)		43 (1.1%)		179 (4%)
Hotel	-	16 (3.5%)		42 (1.1%)		58 (1.3%)
Prison	-	8 (1.7%)		96 (2.4%)		104 (2.3%)
OUTSIDE	5 (12.8%)	31 (6.7%)		1936 (48.9%)		1972 (44.2%)
Binaries	-	-		212 (5.4%)		212 (4.8%)
Street	2 (5%)	5 (1.1%)		445 (11.2%)		452 (10.1%)
Green/park/field	1 (2.5%)	12 (2.6%)		112 (2.8%)		125 (2.8%)
River/Lake	-	4 (0.9%)		10 (0.3%)		14 (0.3%)
Public Area	2 (5%)	10 (2.2%)		1157 (29.2%)		1169 (26.2%)

## Data Availability

All the data have been reported in the manuscript.

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
