# Peer review of "Is It Correct to Consider Caustic Ingestion as a Nonviolent Method of Suicide? A Retrospective Analysis and Psychological Considerations"

_ijerph, 2023, doi:10.3390/ijerph20136270_

Round 1

Reviewer 1 Report

The manuscript was an interesting with valuable data, but some points need to be revised.

1.         TitleThe title is vague, and the authors need to modify it to accurately convey the contents of the manuscript.

2.         AbstractCould you write the data used in this study? In addition, please write the number of subjects or selection of subjects because the number of analyzed cases is very low as a suicide number of a country Moreover, please write about statistical analysis methods used.

3.         Conclusions in AbstractIt is not uncertain why the authors led to this conclusion only when reading this Abstract. Please elaborate more. In addition, please write about results with some figures. Same for conclusions in the main body.

4.         Lines 32shown

5.         Lines 44The sentence is bizarre.

6.         IntroductionIf you want to survey whether caustic ingestion is a kind of violent suicide or not, why didn’t you include and analyze the existing violent suicide in this study?

7.         MethodsPlease add a flowchart of selecting the subjects.

8.         TablesIt might be better to delete unnecessary lines. In addition, you need to accurately describe table titles in order to convey the contents of tables.

9.         Lines 252-254You need to explain more about characteristics of violent suicides in previous studies. In addition, it might be true that characteristics of violent suicides and suicide by caustic ingestion are similar, but does it directly mean that suicide by caustic ingestion should be categorized into violent suicides?

Author Response

The manuscript was an interesting with valuable data, but some points need to be revised.

  1. (Title)The title is vague, and the authors need to modify it to accurately convey the contents of the manuscript.

The title has been changed based on your suggestions.

  1. (Abstract)Could you write the data used in this study? In addition, please write the number of subjects or selection of subjects because the number of analyzed cases is very low as a suicide number of a country Moreover, please write about statistical analysis methods used.

The abstract has been improved as requested. The numbers do not refer to Italy as a whole but only to Milan (a city in the north Italy).

  1. (Conclusions in Abstract)It is not uncertain why the authors led to this conclusion only when reading this Abstract. Please elaborate more. In addition, please write about results with some figures. Same for conclusions in the main body.

Thank you for your comment. Conclusions have been improved and changed acccording to new results. We have also added results in the abstract.

  1. (Lines 32)shown

This word was part of a paragraph that was removed.

  1. (Lines 44)The sentence is bizarre.

We agree and deleted the sentence.

  1. (Introduction)If you want to survey whether caustic ingestion is a kind of violent suicide or not, why didn’t you include and analyze the existing violent suicide in this study?

We added this analysis to the paper.

  1. (Methods)Please add a flowchart of selecting the subjects.

We have added a flowchart as requested.

  1. (Tables)It might be better to delete unnecessary lines. In addition, you need to accurately describe table titles in order to convey the contents of tables.

We agree with the reviewer and have corrected some inaccuracies.

  1. (Lines 252-254)You need to explain more about characteristics of violent suicides in previous studies. In addition, it might be true that characteristics of violent suicides and suicide by caustic ingestion are similar, but does it directly mean that suicide by caustic ingestion should be categorized into violent suicides?

We agree and have added more information on violent suicides.

Reviewer 2 Report

This is a well-written article on the comparison between people that committed suicide using caustic substances and people that committed suicide by chemical ingestion. I have some comments aimed at improving the manuscript.

-        It is not justified why focus the study in Italy, and in Milan in particular. Which is the situation in Italy and Milan regarding suicide, and in particular suicide by caustic substances and chemical ingestion? I suggest that this is addressed in the introduction to better contextualize and define the research problem.

-        In the introduction, the authors dedicate a large portion to the genetic risk of suicide and serotonergic system, focusing also on psychological vulnerability. This part is separated from the rest of the document that focus on suicide methods, and this connection is not addressed in the document.

-        It is important to highlight that the authors criticize that the literature treats nonviolent methods as a homogeneous cluster and that, among the toxic substances that could be ingested, some might be more dangerous than others. However, they only separate caustic substances from the other nonviolent methods, also treating the other methods as a homogeneous cluster. I suggest that, if possible the authors could compare caustic substances not only with the other non-violent methods but also with the violent methods to see if the characteristics of people who committed suicide are similar to the ones from people that used violent methods.

-        The manuscript has a very limited method with which the authors are trying to relate or compare the sociodemographic characteristics of one type of suicide with another. They only use basic association measures, which are too limited and basic. I suggest that they use a more robust method, at least a logistic regression model. If the number of cases is too small, then the authors should consider widening the sample to other cities or states in Italy.

-        Also in the methodology it is not explained how the authors grouped the sociodemographic characteristics.

-        The reasons on why to compare the group of suicides by ingestion of caustics and by chemical exposure is not clear and not well justified. Why not compare the characteristics of ingestion of caustics with violent suicides? As this is one of the main conclusions from the authors, that ingestion of caustics should be treated as a violent suicide, so it would make more sense to compare them to the group they are trying to include them. At least both comparisons should be made.  

-        The results in line 140 are statistically significant, but just barely, which could be related to the differences in group size. Besides, the analysis was made without controlling the effect of the other variables, which is too limited and too descriptive. I strongly suggest a regression analysis to improve the quality of the paper.

-        In lines 148 to 150, the authors mention that group 1 died mostly of organic injuries which directly results from the type of substance used in the suicide. This result is rather expected and does not add too much to the possible discussion.

-        Also, the authors later in the paper argue that these methods (group 1) have a higher lethality than the other group (2), but that result is not obtained from the data from this paper. But they use it as an argument on why the group 1 suicides should be included as a violent type. I suggest the authors discuss only the results that come from the data that they analyzed. This is represented in lies 225 to 227, which are results not seen in this article.   

-        In lines 232 to 243, the authors mention the effects of caustic ingestion in the body. However, this is not related to the objectives of the paper or with the data that was analyzed so I recommend that this part is removed from the manuscript because it should not be discussed as this part should be dedicated to discuss the results from the study with the existent literature and results from other studies.

-        In lines 251 to 254, the authors mention that the ingestion of caustic substances has one of the highest death rates and their characteristics are more similar to violent deaths. However, both results are not obtained in this manuscript. Moreover, the group1 and group2 characteristics only differ by age group and psychiatric disorders (but only in a simple association, not taking into account the other characteristics, which could be achieved using other more complex statistical methods). This argument from the authors favors my suggestion that the paper should also include a comparison of group1 characteristics with violent suicides as a group3, for example.

Author Response

This is a well-written article on the comparison between people that committed suicide using caustic substances and people that committed suicide by chemical ingestion. I have some comments aimed at improving the manuscript.

-        It is not justified why focus the study in Italy, and in Milan in particular. Which is the situation in Italy and Milan regarding suicide, and in particular suicide by caustic substances and chemical ingestion? I suggest that this is addressed in the introduction to better contextualize and define the research problem.

We concurred with the reviewer and included more details regarding this issue.

-        In the introduction, the authors dedicate a large portion to the genetic risk of suicide and serotonergic system, focusing also on psychological vulnerability. This part is separated from the rest of the document that focus on suicide methods, and this connection is not addressed in the document.

We agree with the reviewers that this digression is not relevant to the paper. We have removed this part from the text.

-        It is important to highlight that the authors criticize that the literature treats nonviolent methods as a homogeneous cluster and that, among the toxic substances that could be ingested, some might be more dangerous than others. However, they only separate caustic substances from the other nonviolent methods, also treating the other methods as a homogeneous cluster. I suggest that, if possible the authors could compare caustic substances not only with the other non-violent methods but also with the violent methods to see if the characteristics of people who committed suicide are similar to the ones from people that used violent methods.

As suggested, we made this comparison and added the results obtained.

-        The manuscript has a very limited method with which the authors are trying to relate or compare the sociodemographic characteristics of one type of suicide with another. They only use basic association measures, which are too limited and basic. I suggest that they use a more robust method, at least a logistic regression model. If the number of cases is too small, then the authors should consider widening the sample to other cities or states in Italy.

We added logistic regression models.

-        Also in the methodology it is not explained how the authors grouped the sociodemographic characteristics.

We agree with the reviewers and have added specifics on the ethnic division of the sample.

-        The reasons on why to compare the group of suicides by ingestion of caustics and by chemical exposure is not clear and not well justified. Why not compare the characteristics of ingestion of caustics with violent suicides? As this is one of the main conclusions from the authors, that ingestion of caustics should be treated as a violent suicide, so it would make more sense to compare them to the group they are trying to include them. At least both comparisons should be made.  

We agree with the reviewers and have carried out further analysis to investigate this issue.

-        The results in line 140 are statistically significant, but just barely, which could be related to the differences in group size. Besides, the analysis was made without controlling the effect of the other variables, which is too limited and too descriptive. I strongly suggest a regression analysis to improve the quality of the paper.

We added logistic regression models.

-        In lines 148 to 150, the authors mention that group 1 died mostly of organic injuries which directly results from the type of substance used in the suicide. This result is rather expected and does not add too much to the possible discussion.

We agree with the reviewer and we added “as expected” ad the beginning of the sentence.

-        Also, the authors later in the paper argue that these methods (group 1) have a higher lethality than the other group (2), but that result is not obtained from the data from this paper. But they use it as an argument on why the group 1 suicides should be included as a violent type. I suggest the authors discuss only the results that come from the data that they analyzed. This is represented in lies 225 to 227, which are results not seen in this article.   

We agree with the reviewer and therefore we decided to delete this part from the paper.

-        In lines 232 to 243, the authors mention the effects of caustic ingestion in the body. However, this is not related to the objectives of the paper or with the data that was analyzed so I recommend that this part is removed from the manuscript because it should not be discussed as this part should be dedicated to discuss the results from the study with the existent literature and results from other studies.

We agree with the reviewer and we removed this part.

-        In lines 251 to 254, the authors mention that the ingestion of caustic substances has one of the highest death rates and their characteristics are more similar to violent deaths. However, both results are not obtained in this manuscript. Moreover, the group1 and group2 characteristics only differ by age group and psychiatric disorders (but only in a simple association, not taking into account the other characteristics, which could be achieved using other more complex statistical methods). This argument from the authors favors my suggestion that the paper should also include a comparison of group1 characteristics with violent suicides as a group3, for example.

As suggested from the reviewer, we added this analysis to the paper.

Round 2

Reviewer 1 Report

Thank you for the revision and adding analyses.

1.         (Tables) It is better to equalize font sizes in each table.

2.         Regarding logistic analysis, it is better to write whether it is a multivariate logistic analysis or univariate logistic analysis.

3.         (Methods) Weren’t there missing values in the data used?

4.         (Table 3,5, and 7) It is better to arrange the Table because which groups were compared by the U test are not clear from the Tables.

Author Response

  1. (Tables) It is better to equalize font sizes in each table.

Reply:

Thank you. We equalized the font sizes of all the tables.

  1. Regarding logistic analysis, it is better to write whether it is a multivariate logistic analysis or univariate logistic analysis.

Reply:

We better specify this point (univariate logistic analysis).

  1. (Methods) Weren’t there missing values in the data used?

Reply:

We did not include missing values in the analyses.

  1. (Table 3,5, and 7) It is better to arrange the Table because which groups were compared by the U test are not clear from the Tables.

Reply:

We completely agree. We added this detail in each table.

Reviewer 2 Report

The authors addressed most of the comments I made in my last review. Still, I have some minor concerns regarding the logistic regressions. The results from these analyses are not presented, so I suggest the authors include them in the manuscript. The authors do not report a correlation analysis between the independent variables of the regression model, and thus we don't know if it has problems of multicollinearity, something I suggest is addressed. And finally, there is no mention of the goodness-of-fit of the models, and not tests analyzing this were presented. Therefore, I suggest this tests are estimated and the results shown in the final manuscript.

Author Response

The authors addressed most of the comments I made in my last review. Still, I have some minor concerns regarding the logistic regressions. The results from these analyses are not presented, so I suggest the authors include them in the manuscript.

Reply:

Thank you. We included 2 further tables.

The authors do not report a correlation analysis between the independent variables of the regression model, and thus we don't know if it has problems of multicollinearity, something I suggest is addressed.

Reply:

We reported results related to multicollinearity.

And finally, there is no mention of the goodness-of-fit of the models, and not tests analyzing this were presented. Therefore, I suggest this tests are estimated and the results shown in the final manuscript.

Reply:

We reported the goodness-of-fit of the models.